# Magnitudes of post-abortion family planning utilization and associated factors among women who seek abortion service in Bahir Dar Town health facilities, Northwest Ethiopia, facility-based cross-sectional study

**Amsalu Muchie[1], Fentie Ambaw Getahun[2], Yibeltal Alemu Bekele[3]***, Tsion Samual[4], Tebkew Shibabaw[4]**

**1** Felege Hiwot Referral Hospital, Bahir Dar, Ethiopia, **2** School of Public Health, Bahir Dar University, Bahir Dar, Ethiopia, **3** Department of Reproductive Health and Population Studies, Bahir Dar University, Bahir Dar, Ethiopia, **4** Department of Environmental Health, Bahir Dar University, Bahir Dar, Ethiopia

* yibeltalalemu6@gmail.com

## Abstract

### Introduction

Globally an estimated 55.9 million abortions occur each year. The majority of abortions occur due to unintended pregnancies, which is a result of the non-use of family planning methods. World health organization recommends all clients to utilize modern contraceptive methods after any abortion procedure. However, post-abortion family planning utilization is still low in Ethiopia including the study area. Therefore, this study was expected to determine the utilization of post-abortion family planning and associated factors in Bahir Dar city health facilities in Northwest Ethiopia.

### Method

Institution based cross-sectional study was conducted among 408 women from March 1 to April 30, 2019. Data were collected through face-to-face interview using a structured and pre-tested questionnaire. Systematic random sampling was used to select the study participants. Data were cleaned, coded, and entered into epi data and exported to SPSS for further analysis. Both bivariable and multivariable logistic regression were employed. Those variables that had a p-value of less than 0.2 during the bivariate analysis were retained for the multivariable analysis. P-value and confidence interval were used to measure the level of significance on multi-variable analysis and those variables whose P-value, less than 0.05 was considered as statistically significant.

### Results

The finding of this study showed that the magnitude of post-abortion family planning (PAFP) utilization was 61% with 95% CI (55, 65). Secondary education level(AOR, 4.58; 95% CI

**Data Availability Statement:** All relevant data are within the manuscript and its Supporting Information files.

**Funding:** Bahir Dar University has funded this research. The funders had no role in study design, data collection and analysis, decision to publish, or preparation of the manuscript.

**Competing interests:** The authors have declared that no competing interests exist.

(1.96, 10.69)), certificate and above education level (AOR, 3.06; 95% CI (1.32, 7.08)), Manual Vacuum Aspiration(MVA) (AOR, 7.05; 95% CI (2.94, 16.90)), both medication and Manual Vacuum Aspiration (AOR, 5.34; 95% CI (2.56, 11.13)) and received Post Abortion Family Planning (PAFP) counseling (AOR, 5.99; 95% CI (3.23, 11.18)) were significantly associated with PAFP utilization.

## Conclusion

Post-abortion family planning utilization in Bahir Dar health facilities was low compared with the national figure. Secondary and above educational level, respondents who were managed by Manual Vacuum Aspiration (MVA), both Manual Vacuum Aspiration (MVA) and medication and receiving Post Abortion Family Planning (PAFP) counseling were predictors of post-abortion family planning service utilization.

## Introduction

Post-abortion family planning(PAFP) is the initiation and use of family planning methods immediately after an abortion [1]. It is the part of abortion care service which increases contraceptive use prevalence and reduces unintended pregnancies and unsafe abortion [2]. The WHO guideline recommends a woman to wait at least six months after an abortion before getting pregnancy. This is important for her body time to regain its strength and prepare for healthy pregnancy [3].

Globally an estimated 55.9 million abortions occur each year. The majority (49.3 million) abortions occur in developing countries [4]. Unsafe abortion is preventable but it still causes 13% of maternal deaths and 20% of the overall burden of maternal death and long-term disability [13]. In Ethiopia, an estimated 620,300 induced abortions were performed in 2014 with an annual abortion rate of 28 per 1,000 women [5].

Studies conducted in developing countries showed that the prevalence of post abortion family planning utilization were 73%, 73%, 81%, and 97.7% in Africa and Asia, Pakistan, India and Brazil respectively [6–9]. Similar studies conducted in Sub-Saharan countries showed that the prevalence of post abortion family planning utilization ranges from 55.7% to 76% [10, 11]. Previous studies conducted in different parts of Ethiopia also showed the prevalence of post abortion family planning utilization were 47.5%, 56.5%, 59.2%, and 70% in Dessie Town, Guragie Zone, Debre Markos Town, and Jimma Town respectively [12–14].

According to different studies conducted in the developing countries, age, marital status, educational level, type of health facility, level of health facility, method mix, sex preference of health care provider, the desire of having more children, parity, gravidity, partner refusal, women accompanied by their partner, previous live birth, fear of side effects, and lack of adequate information determined post-abortion family planning utilization [12, 15–21]. Similarly, different studies in Ethiopia showed age, marital status, fear of side effect, availability of the contraceptive method, previous history of abortion, knowledge about family planning, misconception about contraceptive determined post-abortion family planning utilization [22–24].

Currently, the government of Ethiopia is committed to achieving the Sustainable Development Goal 3 (SDG-3) that promise to end all preventable causes of maternal death through providing effective maternal health care services [25, 26]. It is recognized that providing post-abortion family planning services for women minimize the risk of pregnancy-related problems. In addition, the government of Ethiopia implemented the new health sector transformation plan

(HSTP) that aimed to improve the uptakes of maternal health care services utilization. One of the focus areas is expanding the infrastructure for providing post abortion family planning services since 2015/16 [21]. However, there is paucity of evidence that indicates the magnitude of post-abortion family planning services and its associated factors in Ethiopia after the implementation of the HSTP including the study area. So the aim of this study was to determine the magnitude and associated factors of post-abortion family planning service.

## Methods

### Study area and design

This study was carried out in Bahir Dar town, North West Ethiopia. Bahir Dar is the capital city of the Amhara regional state. It is located around 565 km far from the Ethiopian capital, Addis Ababa. In the city, there are two government hospitals, five government health centers, two higher privet clinics, and one private reproductive health-based clinic that are providing abortion care services. Institutional based cross-sectional study was conducted from March 1 to April 30, 2019. All women who were seeking abortion care service in Bahir Dar city health facilities were the source population. All women who came for abortion care services during the study period were included in this study while women who were critically ill and showed any sign of infection excluded in this study.

### Sample size and procedure

The sample size was calculated using a single population proportion formula by considering the following assumption; according to a study conducted in Debre Markos Town, the proportions of post-abortion family planning service utilization was 59.2% [27]. With a 95% confidence interval, 5% margin of error, and 10% none response rate. Therefore, a total of 408 women who received abortion care services were included in this study.

All public and private health institutions (a total 10 governmental health facilities and 4 private health facilities) which are providing abortion care service found in the town were selected. All selected health care facilities provided both legal abortion services based on the country law and emergency post abortion care services for those who need the service. The participants were proportionally allocated to each health facility based on last year's client flow of abortion care services performance. Finally, systematic random sampling was used to select the study participants in each health care facility.

### Study variables

**Dependent variables.** Post-abortion family planning utilization (Yes/No).

**Independent variables.** *Socio-demographic related factors.* Age, monthly income, religion, marital status, occupation, family support, and educational status

*Reproductive and abortion management-related factors.* gravidity, parity, the desire of children, previous abortion, surgical abortion, medical abortion, and gestational age.

*Facility and provider related factors.* Type of health institution, level of health institutions, PAFP counseling, availability of the contraceptive, sex of the provider, method mix, and communication.

*Personal and family planning method factors.* Knowledge, request for service, information about family planning methods, and family planning side effects.

### Operational definition

**Knowledgeable.** Seven questions were used to measure knowledge about post-abortion modern contraceptive uses. Respondents who answered the mean and above among seven

knowledge related family planning questions were considered as "knowledgeable" about post-abortion modern contraceptive uses while respondents who answered below the mean knowledge related family planning questions were considered as "not knowledgeable" [20].

### Data collection procedure and tools

Data were collected through a structured questionnaire developed by reviewed different literature vie interviews. The questionnaire was first prepared in English and translated to local language Amharic and then translated back to English to check the consistency. The questionnaire was comprised of socio-demographic characteristics, individual-related characteristics, reproductive health-related and facility-related characteristics. Five BSc nurses were assigned for data collection and one MSc holder nurse was assigned to supervise during the whole data collection process.

### Data quality control

Pre-test was conducted on 10% [41] respondents in Finote Selam Hospital. One day training was provided for data collectors and the supervisor on the objective, and the relevance of the study. The supervisor managed the data collection process every day, and the principal investigator also checked the completeness of the questionnaire every day.

### Data processing and management

Data were cleaned, coded, and entered into Epi data version 3.1 and exported to SPSS version 23 for further analysis. Descriptive statistics were carried out to see the distributions of independent variables. Both bivariable and multivariable logistic regression analysis were employed. On bivariable analysis p-value of less than 0.2 was used to select candidate variables for multivariable analysis. P-value and 95% confidence interval were used to measure the level of significance on multivariable analysis and those variables with a P-value of less than 0.05 on multivariable analysis were considered as statistically significant.

### Ethical statement

Ethical clearance was obtained from the Ethical review board of Bahir Dar University, College of Medicine and Health Sciences. A support letter was received from Amhara public health institute and Bahir Dar city administration health office.

Written consent was taken for every participant and based on their agreement the data collection took place. Consent was taken from their guardian for women whose age less than 18 years. Information was provided for all participants about the objective, the purpose and the contents of the study as well as their rights to refusal at any time of data collection. The participants were also reassured how to handling and uses of the data.

## Result

### Socio-demographic characteristics

A total of 408 respondents participated in this study with a response rate of 100%. One hundred fifty-six (38.2%) respondents were in the age group of 25 to 29 with the mean age of 26.23 and SD of ±4.75. Two hundred forty (58.8%) respondents were married. Three hundred thirty-six (82.6%) respondents were Orthodox religion followers and 72(17.4%) respondents were Muslims. Three hundred ninety (95.6%) respondents were from Amhara and 18(4.4%) were from Agew ethnic groups. Eighty-four (20.6%) respondents did not attend formal

education while 118 (28.9%) respondents attended secondary education. Three hundred fifteen (77.2%) respondents were living in rural areas (Table 1).

## Reproductive hearth related characteristics

One hundred eighty-three (44.8%) respondents have prim-gravidity. Thirty-six (8.8%) respondents have had a history of previous abortion. Three hundred thirty eighty (82.8%) respondents desire to have additional children for the future. Two hundred forty-two (59.3%) respondents come with spontaneous abortion. Two hundred eighty-three (69.4%) respondents received post-abortion family planning counseling before they left the health institution and 144 (35.3%) of them got individual counseling (Table 2).

## Magnitudes of post abortion family planning utilization

The magnitude of post abortion family planning utilization in the study area was 249(61%) with 95% (CI: 55.0, 65.0). Two hundred eighty-two (69%) respondents were received abortion

**Table 1. Socio-demographic characteristic of respondents in Bahir Dar town, Amhara region, Northwest Ethiopia, 2019.**

| Variables (n 408) | Frequency | Percent (%) |
|---|---|---|
| **Age** | | |
| 15–19 | 19 | 4.7 |
| 20–24 | 135 | 33.1 |
| 25–29 | 156 | 38.2 |
| 30–34 | 61 | 15 |
| $\geq 35$ | 37 | 9.1 |
| **Marital status** | | |
| Single | 141 | 34.5 |
| Married | 240 | 58.8 |
| Other (widowed and divorced) | 27 | 6.6 |
| **Level of Education** | | |
| Illiterate | 84 | 20.6 |
| Primary education | 89 | 21.8 |
| Secondary education | 118 | 28.9 |
| Certificate and above | 117 | 28.7 |
| **Occupation** | | |
| Housewife | 92 | 22.5 |
| Farmer | 46 | 11.3 |
| Merchant | 94 | 23 |
| Governmental Employee | 40 | 9.8 |
| Daily Laborer | 36 | 8.8 |
| Student | 100 | 24.5 |
| **Residence** | | |
| Urban | 93 | 22.8 |
| Rural | 315 | 77.2 |
| **Monthly Income** | | |
| <1000 | 51 | 12.5 |
| 1001–2000 | 49 | 12 |
| >2001 | 308 | 75.5 |

**Table 2. Reproductive health related characteristics of respondents in Bahir Dar town, Amhara Region, north-west Ethiopia, 2019.**

| Variable | Frequency | Percent (%) |
|---|---|---|
| **Type of abortion management** | | |
| Medication | 181 | 44.3 |
| MVA | 67 | 16.4 |
| Both | 160 | 39.2 |
| **Type of health institution** | | |
| Public | 248 | 60.8 |
| Private | 160 | 39.2 |
| **Want additional child** | | |
| Yes | 338 | 82.8 |
| No | 70 | 17.2 |
| **Planned Pregnancy** | | |
| Yes | 222 | 54.4 |
| No | 186 | 45.6 |
| **Know about PAFP method** | | |
| Yes | 172 | 42.2 |
| No | 236 | 57.8 |
| **Source of information** | | |
| Neighbor | 108 | 26.5 |
| Husband | 14 | 3.4 |
| Provider | 159 | 39 |
| Mass media | 24 | 5.9 |
| **Post-abortion counseling** | | |
| Yes | 283 | 69.4 |
| No | 125 | 30.6 |
| **Counseling type** | | |
| Individual | 132 | 46.6 |
| With husband | 111 | 39.2 |
| With parents | 40 | 14.1 |
| **Knowledge about PAFP** | | |
| Knowledgeable | 180 | 44 |
| Not Knowledgeable | 228 | 56 |
| **Sex preference of the health care provider** | | |
| Male | 162 | 39.7 |
| Female | 176 | 43.1 |
| Both | 70 | 17.2 |

service at hospital level while 127(31%) respondents were received the service at the health centers.

## Factors of post-abortion family planning utilization

In bivariable analysis, age of the women, educational status, marital status, residence, gravidity, parity, the desire of additional children, previous history of abortion, gestational age of current pregnancy, sex preference of health care provider, type of abortion management, communication skill, accompanied with a husband, type of health institution, level of the health institution, and PAFP counseling were found to be candidate variables at P-value < 0.2 for the multivariable analysis. On multivariate analysis, educational status, post-abortion family

**Table 3. Factors associated with post abortion modern contraceptive utilization among women who came for abortion service in Bahir Dar city health institutions, North West Ethiopia, 2019.**

| Variable | PAFP utilization | | COR (95%CI) | AOR (95% CI) |
|---|---|---|---|---|
| | Yes | No | | |
| **Educational status of the mother** | | | | |
| Not attained formal education | 30 | 54 | 1 | 1 |
| Primary education | 39 | 50 | 1.40(0.76,2.58) | 0.95(0.40,2.25) |
| Secondary education | 86 | 32 | 4.83(2.64,8.84) | 4.58(1.69,10.69 |
| Tertiary education | 91 | 26 | 6.30(3.77,11.75) | 3.06(1.32,7.08) |
| **Receive PAFP Counseling** | | | | |
| Yes | 170 | 46 | 5.64(3.65,8.72) | 5.99(3.20,11.18) |
| No | 76 | 116 | 1 | 1 |
| **Abortion Management** | | | | |
| **Procedure** | | | | |
| Medication | 69 | 112 | 1 | 1 |
| MVA | 53 | 14 | 6.14(3.17,11.00) | 4.76(1.93,11.76) |
| Both | 124 | 36 | 5.59(3.47,9.01) | 4.62(2.18,9.81) |
| **Number of pregnancies** | | | | |
| One | 85 | 100 | 1 | 1 |
| Two | 68 | 28 | 2.85(1.68,4.83) | 2.22(0.99,4.97) |
| Three | 46 | 12 | 4.51(2.24,9.06) | 2.94(0.42,7.74) |
| Four | 47 | 22 | 2.51(1.40,4.50) | 0.76(0.29,2.00) |
| **Type of institution** | | | | |
| Public | 167 | 81 | 2.11(1.40,3.18) | 1.64(0.89,3.00) |
| Private | 79 | 81 | 1 | 1 |
| **Number of children** | | | | |
| No child | 99 | 103 | 1 | 1 |
| One | 69 | 29 | 2.47(1.48,4.14) | 0.15(0.01,1.59) |
| Two | 37 | 12 | 3.20(1.58,6.50) | 0.65(0.09,4.56) |
| Three and above | 41 | 18 | 2.37(1.27,4.40) | 6.90(0.27,172.7) |
| **Previous abortion** | | | | |
| Yes | 37 | 9 | 3.02(1.41,6.45) | 3.54(0.90,13.90) |
| No | 208 | 154 | 1 | 1 |

planning counseling, and type of abortion management were factors that affect post-abortion family planning utilization at p-value less than 0.05.

The odds of post-abortion family planning utilization among women who attended secondary educational level were 4.58 times higher than those who did not attend formal education [AOR = 4.58 (95%CI:1.69,10.69)]. The odds of post-abortion family planning utilization among women who attended a certificate and above educational level were 3.06 times higher than those who did not attend formal education [AOR = 3.06 (95%CI:1.32, 7.08)]. The odds of post-abortion family planning utilization among women who received post abortion family planning counseling were 5.99 times higher than those who did not received [AOR = 5.99(95%CI:3.20,11.18)].

The odds of post-abortion family planning utilization among women who managed by MVA was 4.76 times higher than those who managed by medication [AOR = 4.76 (95% CI: 1.93, 11.76)]. Similarly, the odds of post-abortion family planning utilization among women whose abortion procedure managed by both MVA and medication were 4.62 times higher than those who managed by medication alone [AOR = 4.62 (95%CI: 2.18, 9.81)] (Table 3).

## Discussion

In this study, the overall post-abortion family planning utilization was 61% with 95% (CI: 55.0, 65.0). The finding of this study was consistent with previous studies conducted in Guragie district and Debire Markos Town, with overall post-abortion family planning utilization of 56.5% [28] and 59.2% [25] respectively. However, the finding of this study was lower than the previous studies conducted in Addis Ababa [26], Ethiopia [14] and Brazil [29]. This variation may be due to marital status differences. The percentage of married women was higher in the study from Brazil (86.7%) compared to our finding in which 58.8% of the respondents were married. Presence of more married women may increase the utilization of PAFP since they are more autonomous to decide family planning use [26]. The other reason may be due to educational level differences. A study conducted in Addis Ababa showed that, 75% of respondents attended secondary and above educational levels [25] but the current study showed only 57.6% of respondents were secondary and above educational levels accomplishers. In Brazil, 92.6% of respondents were knowledgeable which may increase the utilization of post-abortion modern contraceptive [30]. In addition, the finding of this study was higher compared to the previous study done in Dessie Town (47.5%). This difference might be due to the difference in the number of respondents who counseled about post abortion family planning. A study conducted in Dessie Town showed only 56% of respondents received post-abortion family planning counseling [31] while in the current study, 69% of the respondents received post-abortion family planning counseling [15].

The odds of post-abortion family planning use among women who attended secondary education and above was higher compared to women who did not attend formal education. The finding of this study was similar to the previous studies conducted in Debre Markose Town [32], Gambella [33], and Kenya [34]. This might be due to the reason that educated women have better access to information about family planning and reproductive health issues [35] which enables them to pass informed decisions and more concerned about their reproductive health rights [14]. This implies that extending women's education to at least secondary level increases the uptakes of PAFP contributing to achieving SDG 3.1 to end all preventable causes of maternal death.

The odds of post-abortion family planning use among women who received post-abortion family planning counseling was higher than the counterparts. The finding of this study was similar with the previous studies conducted in Debre Markos [36], Kenya [37] and Brazil [38]. This might be due to counseling helps the women to make informed decisions about family planning services utilization [39]. Counseling is one of the critical elements in the provision of quality family planning services. It is a means, provider help clients make and carry out their own choices about reproductive health and family planning [40]. Post-abortion contraceptive counseling is an effective way of increasing the utilization of highly effective methods of contraception [41].

The odds of post-abortion family planning utilization among women whose abortion procedure managed by MVA was higher compared to women managed by medication alone. This finding was in line with the previous studies conducted in England and Wales [42] and Pakistan [43]. However, the finding of this study was different from the study conducted in India [44]. This discrepancy might be due to variation in women preference to nature of procedure, which takes a shorter time [45]. Similarly, the odds of post-abortion family planning use among women managed with both (manual vacuum aspiration and medication) were higher compared to women managed with medication. This finding was supported by the study conducted in eight countries of Africa [46].

## Conclusions

The magnitude of post-abortion family planning utilization in Bahir Dar city health facilities was low compared to the national figure. Attending education, type of abortion management and receiving counseling were predictors of post-abortion family planning method utilization. Based on the findings, we recommend that making more efforts on counseling service provision by Ministry of Health (MoH) is the effective and sustainable method in increasing PAFP utilization of women.

## Supporting information

**S1 File. Data set, that contain the survey data of the participant.**
(XLSX)

## Acknowledgments

First of all, the authors would like thanks to all participants who were volunteer to participate in this study. We are also grateful to the Bahir Dar town health office staff, health professionals for their invaluable support through the whole process and the supervisors and data collectors who have committed themselves throughout the study period.

 **Declaration:** *Consent for publication*. Verbal consent for publication was received from the participant with regard to all the detail that explains the participants.

## Author Contributions

**Conceptualization:** Amsalu Muchie, Fentie Ambaw Getahun, Yibeltal Alemu Bekele.

**Data curation:** Amsalu Muchie, Fentie Ambaw Getahun, Yibeltal Alemu Bekele.

**Formal analysis:** Amsalu Muchie, Fentie Ambaw Getahun, Yibeltal Alemu Bekele.

**Investigation:** Amsalu Muchie, Fentie Ambaw Getahun, Yibeltal Alemu Bekele.

**Methodology:** Amsalu Muchie, Fentie Ambaw Getahun, Yibeltal Alemu Bekele.

**Software:** Amsalu Muchie, Fentie Ambaw Getahun, Yibeltal Alemu Bekele.

**Supervision:** Amsalu Muchie, Fentie Ambaw Getahun, Yibeltal Alemu Bekele.

**Validation:** Amsalu Muchie, Fentie Ambaw Getahun, Yibeltal Alemu Bekele.

**Writing – original draft:** Amsalu Muchie, Fentie Ambaw Getahun, Yibeltal Alemu Bekele.

**Writing – review & editing:** Amsalu Muchie, Fentie Ambaw Getahun, Yibeltal Alemu Bekele, Tsion Samual, Tebkew Shibabaw.

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
