## [Decision Letter · Decision Letter 0]

2 Oct 2020

PONE-D-20-26289

Magnitudes of post-abortion family planning utilization and associated factors among women who seek abortion service in Bahir Dar Town health facilities, Northwest Ethiopia, Facility-Based Cross-Sectional Study.

PLOS ONE

Dear Dr. Bekele,

Thank you for submitting your manuscript to PLOS ONE. After careful consideration, we feel that it has merit but does not fully meet PLOS ONE’s publication criteria as it currently stands. Therefore, we invite you to submit a revised version of the manuscript that addresses the points raised during the review process.

Four experts in the field handled your manuscript. Although they found some interest in your study, several major comments arose during review. ALL of the reviewers' comments need to be addressed in your revised manuscript.

We look forward to receiving your revised manuscript.

Kind regards,

Frank T. Spradley

Academic Editor

PLOS ONE

2. In your Methods section, please provide additional information about the participant recruitment method and the demographic details of your participants.

Please ensure you have provided sufficient details to replicate the analyses such as:   

-    a description of any inclusion/exclusion criteria that were applied to participant recruitment,

-    a statement as to whether your sample can be considered representative of a larger population,

-    a description of how participants were recruited, and

-       descriptions of where participants were recruited and where the research took place.

3. Please include additional information regarding the survey or questionnaire used in the study and ensure that you have provided sufficient details that others could replicate the analyses.

For instance, if you developed a questionnaire as part of this study and it is not under a copyright more restrictive than CC-BY, please include a copy, in both the original language and English, as Supporting Information. 

If the original language is written in non-Latin characters, for example Amharic, Chinese, or Korean, please use a file format that ensures these characters are visible.

4. Please provide additional details regarding participant consent.

In the ethics statement in the Methods and online submission information, please ensure that you have specified what type you obtained (for instance, written or verbal, and if verbal, how it was documented and witnessed).

If your study included minors, state whether you obtained consent from parents or guardians.

If the need for consent was waived by the ethics committee, please include this information'

5. Thank you for stating the following above the Acknowledgments Section of your manuscript:

'Funding

Bahir Dar University has funded this research'

'The author(s) received no specific funding for this work.'

Reviewers' comments:

Reviewer's Responses to Questions

**Comments to the Author**

1. Is the manuscript technically sound, and do the data support the conclusions?

Reviewer #1: Partly

Reviewer #2: No

Reviewer #3: Partly

Reviewer #4: Partly

2. Has the statistical analysis been performed appropriately and rigorously? 

Reviewer #1: Yes

Reviewer #2: No

Reviewer #3: Yes

Reviewer #4: Yes

3. Have the authors made all data underlying the findings in their manuscript fully available?

Reviewer #1: Yes

Reviewer #2: Yes

Reviewer #3: Yes

Reviewer #4: No

4. Is the manuscript presented in an intelligible fashion and written in standard English?

Reviewer #1: No

Reviewer #2: No

Reviewer #3: No

Reviewer #4: No

5. Review Comments to the Author

Reviewer #1: Introduction

Para 2: “Globally an estimated 55.9 million abortions occur each year, from this 49.3 million occur in the developing”. This sentence is obscure, please make it clear.

Para 2: What the author meant by writing “long-term disability”?

Para 3: “Studies conducted in developing countries showed that the prevalence of post abortion family planning utilization was 73%, 73%, 81%, and 97.7% in Africa and Asia, Pakistan, India, and Brazil respectively” . Pakistan and India are Asian countries, why do you mention prevalence in Africa and Asia?

The last paragraph: Why do you conduct the same research in this area since there are studies concerning the utilization of family planning service in your country?

Method

The author should provide the data on the health institutions, such as the number of abortions and socioeconomic and obstetric profile of population studied.

I missed one paragraph describing inclusion and exclusion criteria.

It’s not clear the criteria for the inclusion of explanatory variables in the multivariate logistic regression. It should be explained.

Discussion

Again, why do you conduct the same research in this area since there are studies concerning the utilization of family planning service in your country? It is important to compare this result with findings from other studies from Ethiopia and eventually from abroad? The author stated” in the number of respondents who counseled about post abortion family planning” when explaining the difference, this is not convincing.

When the author explain: “The odds of post-abortion family planning use among women who received post-abortion family planning counseling were higher than the counterpart”, you should focus on the benefits which post-abortion family planning counseling might bring to abortion patients.

I missed one paragraph explaining: “sex preference of health care providers, and counseling about post-abortion family planning were predictors of post-abortion family planning method utilization”.

Conclusion

It should be shorter and concise. Recommendations should be more specific and target the risk factors in your research.

Reviewer #2: Understanding the utilization of postabortion family planning is a worthwhile undertaking and critical to reducing unintended pregnancies as well as maternal mortality in places like Ethiopia. Congratulations to the authors for conducting this research and putting together this draft for publication. Some specific feedback to strengthen the paper is listed here:

-Spend more time on the definitions and ensure the use of globally recognized definitions. It's unclear whether the authors are considering contraception after INDUCED abortions or also considering contraception after SPONTANEOUS abortions (miscarriages). Postabortion care generally refers to both and PAFP is important in both cases--and spontaneous abortions are much more common than induced (which means the globally cited numbers should be higher). It would be helpful to define PAC services broadly (treatment of complications, counseling, and provision of contraception) and emphasize that PAFP is an essential component of quality PAC services. Also consider using the global estimate for abortion from Guttmacher or another source.

-What's the population of Bahir Dar town? How many women of reproductive age? What's the contraceptive prevalence rate in the region/area? What's the total fertility rate? What other contextual information and population data can you provide to give a better picture of why understanding PAFP rates in Bahir town is important? What does studying this area contribute to broader learning about PAFP in Ethiopia / Horn of Africa / Globally?

-Why was Debre Markose town used as a proxy to calculate the sample size? What are the similarities? what are the differences? How was this considered?

-How many public and private health institutions provide abortion services in Bahir Dar town? What source of information was used to ensure ALL public and private institutions were included (assuming it's often difficult to enumerate all private providers)? Did all included institutions offer postabortion contraception in addition to abortion services? Did all institutions offer similar method options? Were there differences in PAFP acceptance rates for each facility? Was there any learning here? how did this factor into the statistical analysis?

-The methods section needs clarified. Were you collecting this information from patient register books (register extraction?)? Or were you contacting the women for a survey prospectively as they went in for their abortion? Were women approached in their homes? At the facilities? How was the data collected and protected? How many women were approached? How many women gave consent to participate? How many women refused? Would you expect any differences in those who consented and those who refused? The data quality control "pretest" does not make sense, what was done here?

-ETHICS: Was there any sort of ethical approvals for this study (I see in the declaration, but include in your methods section of the paper)? How did you ensure informed consent to approach these women and include their information? "Written verbal consent" does not make sense. How did you protect their information, given the sensitivity of the subject matter? A 100% response rate is quite impressive, but also raises questions about the consent process.

-In the discussion, PAC-FP research has been conducted in many places. Why choose to compare this data to Brazil? I'm sure there are many differing factors in addition to the marriage rate that should be considered.

-Is there any data on the quality of the PAC-FP counseling provided? The quality of the counseling matters for PAFP uptake.

-I would recommend to re-do the discussion and instead of comparing the results of this study to those conducted in other places far away, instead re-work to discuss what the implications of this data are. How can these findings be used programmatically to improve PAC and PAFP uptake in Bahir Dar? What should the MOH be doing? What should the private facilities be doing? What should change now that we have these data? What's next? So what?

Reviewer #3: The manuscript is riddled with a lot of typos and grammatical errors that could have been easily corrected. It is not clear what the sample size was and with such a large volume of clients sampled it would be important to add an exclusion criteria. The number of respondents is very high for the period of the study (one month). Had the study subjects come for treatment for abortion complications related to incomplete abortion or come for termination of pregnancy/safe abortionm is it legal in Ethiopia? at what stage post treatment were the interviews conducted?

Also, how were the data collectors assigned to the facilities for data collection? Finally would be good to revise the referencing - not clear what and where online the references were accessed or whether these were accessed as hard copies. Finally not clear what new learning the study is contributing and if it is specifc learning for the study area please add how the findinsg are expected to be applied. Globally there is not new evidence the study adds.

Reviewer #4: Thank you for inviting me to review this manuscript. It contributes to the science of postabortion care family planning globally and gives further data points on factors related to postabortion family planning. It is clear the authors and researchers worked diligently on this research.

1. Is the manuscript technically sound, and do the data support the conclusions? For the most part this manuscript is technically sound. The conclusions and discussion could be developed further. More specific recommendations are needed.

2. Has the statistical analysis been performed appropriately and rigorously? Yes, although an additional table with the bivariate analysis results would be useful as would the p values for the multivariate table. Also there is a need to clarify the title of Table 3

3. Have the authors made all data underlying the findings in their manuscript fully available? The information in the manuscript just states the data are available in an Excel form, but not exactly how to access them.

4. Is the manuscript presented in an intelligible fashion and written in standard English? There are a number of grammar, spelling, and proof reading issues in this manuscript that need to be addressed as well as inconsistencies in the use of acronyms.

Additional comments for the authors:

- Clarification of education levels – unclear to global audience what “Certificate and above” indicates for education - is also does not correspond to the levels in the analysis or discussion.

- Third paragraph percentage reports are confusing, recommend putting them in parenthesis next to the places

- Be more specific in aim of the study – include the region/area

- Define what you mean by post abortion family planning utilization

- Give what currency the monthly income is in

- Include information about legality of abortion/postabortion care in Ethiopia

- In the introduction and discussion, previous studies don’t specify exactly where they took place – for example in the first paragraph of the discussion is lists Addis Ababa and then Ethiopia – would be helpful to be more specific with the general Ethiopia information – list like with like places. This is the case also in the third paragraph of the discussion – I assume that Debre Markose and Gambella are in Ethiopia, but would be helpful to clarify this as the next one is about Kenya generally

- The last line of the conclusion does not really summarize the findings of the study or give specific recommendations – this is a broad recommendation that could be difficult to complete – it may be useful to specify more specific interventions – which could be related to education but also to the other variables.

6. PLOS authors have the option to publish the peer review history of their article (what does this mean?). If published, this will include your full peer review and any attached files.

Reviewer #1: No

Reviewer #2: No

Reviewer #3: **Yes: **STEMBILE MUGORE

Reviewer #4: No

---

## [Author Response · Author response to Decision Letter 0]

27 Oct 2020

Point by point responses

 Reviewer Comments 

 General comments Authors response 

 The paper is good but is riddled with typos and could do with thorough editing. Thank you for your suggestion. For addressing the issue we consulted senior public health staff and language professors in my university.

1 While the sample size is clear, it is not clear how the clients for the study were recruited and agreed to participate in the study. Were all the 408 women post abortion care clients – please specify in the methodology section? In addition, please clarify the inclusion and exclusion criteria. Thank you for your comment. Based on your comment we amended it on the revision manuscript. 

Line number 189-191

2 Under method: Please clarify the number of health facilities as 408 post abortion clients in one month seems to be high.

 Thank you for your comment. Based on your recommendation we add the numbers health facilities included in the study on the revision manuscript. 

Line number 198-199

3 Please clarify whether these women had come for abortion services, is it legal or whether they had come for treatment of abortion complications.

 Thank you for your comment. All selected facilities provided legal abortion services based on the country law and post abortion care services for those women who came for emergency post abortion care services. We included this information in method section. 

Line Number 199-201

4 The difference by type of treatment is specified please clarify who the level of the provider (nurse, midwife, doctor etc.) as this has implications on whether the client was counseled or not.

 Thank you for your comment. In Ethiopia there is special training (post abortion care training) gave for the profession who gave abortion service otherwise they are not eligible for providing abortion services whatever the level and the types of health professional. The main aim of the training is to increase the capacity of the professionals to provide all components of abortion care including counseling and providing post abortion family planning care. Otherwise there is no variation on counseling’s of family planning on the level of the provider (nurse, midwife, doctor etc.)

5 Four towns as well as the capital are mentioned as part of the comparison – are these like the study area in terms of whether these are rural, similar client demographics and socio-cultural factors that affect family planning decision making? In other words, did they have the same variables that influence FP uptake? Thank you for your comment. The socio-cultures of the capital (Addis Ababa) is almost the same in the study area Bahir Dar with regards of the availability of facilities like health facilities, educational centers that directly influence the uptakes of family planning services since Bahir Dar is one of the second economic centers of the country. 

6 Difficult to understand what was evidence this study added to what is already known to influence PAFP from Ethiopia itself as well as global. Thank you for your comments. As a country Ethiopia implements a strategy that focus on providing post abortion family planning services for all women who receive abortion services while this study find out, the current levels of the post abortion family planning in study area was low only half of them were used family planning services that is contradict the current implemented strategy. Therefore, the lesson we learned from this study is still the country needs to do great attention in the area. In addition, providing counseling for clients is one of the means to improving the uptakes of post-abortion family planning. Therefore, this finding providing a clue to see the way the strategy was implemented.

 Additional points need clarification 

 Introduction 

1 Para 2: “Globally an estimated 55.9 million abortions occur each year; from this 49.3 million occur in the developing”. This sentence is obscure, please make it clear. Thank you for your comments. Based on your comment we correct it on the updated manuscript.

Line number 8 to 9 

2 Para 2: What the author meant by writing “long-term disability”? Thank you for your comments. As we know abortion responsible for maternal mortality and morbidity. From the morbidity abortion causes “long-term disability.” Long term disability mean scarring of the uterine lining, secondary infertility; ectopic pregnancy, damage to internal organs and breast cancer.

3 Para 3: “Studies conducted in developing countries showed that the prevalence of post abortion family planning utilization was 73%, 73%, 81%, and 97.7% in Africa and Asia, Pakistan, India, and Brazil respectively” . Pakistan and India are Asian countries, why do you mention prevalence in Africa and Asia? Thank you for your comment. There is no specific reason that we include African and Asia. But we are interested to show what the magnitudes post-abortion family planning utilization out sides of the study country because of that we found astudy conducted in eight country found in Africa and Asia (Benson J, Andersen K, Brahmi D, Healy J, Mark A, Ajode A, et al. What contraception do women use after abortion? An analysis of 319,385 cases from eight countries. Global public health. 2018;13(1):35-50.) that is why we include it. 

4 The last paragraph: Why do you conduct the same research in this area since there are studies concerning the utilization of family planning service in your country? 

 Thank you for your comment. based on your comment we incorporate some additional evidence that lead as to conduct this study by including “The governments of Ethiopia implemented the new health sector transformation plan to improve the health system utilization one of the focus areas is expanding the infrastructure and post abortion family planning services since 2015/26.” 

Line number 178 to 183

 Method

5 The author should provide the data on the health institutions, such as the number of abortions and socioeconomic and obstetric profile of population studied. I missed one paragraph describing inclusion and exclusion criteria.

It’s not clear the criteria for the inclusion of explanatory variables in the multivariate logistic regression. It should be explained. Thank you for your comment. Based on your comment we amended it on the revision manuscript by including the inclusion criteria “All women who came for abortion care services during the study period were included in this study while women who critically ill and any sign of infection were not included in this study.”

Line number 189-191

 Discussion

6 Again, why do you conduct the same research in this area since there are studies concerning the utilization of family planning service in your country? It is important to compare this result with findings from other studies from Ethiopia and eventually from abroad? The author stated” in the number of respondents who counseled about post abortion family planning” when explaining the difference, this is not convincing. 

 Thank you for your comment. There is different research conducted in Ethiopia in the area of post abortion care before 2015. But in 2015/16 Ethiopia design and implemented the new Health sector transformation plan since with aim of improving quality and equity, universal health coverage and transformation. One of the main focus areas is expanding the infrastructure of post abortion care and components of services. However there is limited finding with area of post abortion family planning services to understand whether there is improvement in the uptake of post abortion family planning. That why we conduct this research. 

7 When the author explains: “The odds of post-abortion family planning use among women who received post-abortion family planning counseling were higher than the counterpart”, you should focus on the benefits which post-abortion family planning counseling might bring to abortion patients. Thank you for your comment. We incorporate it on the updated manuscript “Counseling is one of the critical elements in the provision of quality family planning services. It is a means, provider’s help clients make and carry out their own choices about reproductive health and family planning.”

Line number 330 to 333

8 I missed one paragraph explaining: “sex preference of health care providers, and counseling about post-abortion family planning were predictors of post-abortion family planning method utilization”. Thank you for your comment. We know sex preference of the professional is one of the significant variables on multivariate analysis. But we are not confortable to discuss the variable since it’s not such important for intervention.

 Conclusion

9 It should be shorter and concise. Recommendations should be more specific and target the risk factors in your research. Thank you for your comment. Based on your comment we correct conclusion and recommendation section on the updated manuscript. 

Line Number 346 to 349

---

## [Decision Letter · Decision Letter 1]

23 Nov 2020

PONE-D-20-26289R1

Magnitudes of post-abortion family planning utilization and associated factors among women who seek abortion service in Bahir Dar Town health facilities, Northwest Ethiopia, Facility-Based Cross-Sectional Study.

PLOS ONE

Dear Dr. Bekele,

Thank you for submitting your manuscript to PLOS ONE. After careful consideration, we feel that it has merit but does not fully meet PLOS ONE’s publication criteria as it currently stands. Therefore, we invite you to submit a revised version of the manuscript that addresses the points raised during the review process.

There are still major concerns about your article. Notably, a professional copyeditor must be contacted to proof your manuscript for typographical errors and standard English. Please address ALL comments in your revised manuscript.

We look forward to receiving your revised manuscript.

Kind regards,

Frank T. Spradley

Academic Editor

PLOS ONE

Reviewers' comments:

Reviewer's Responses to Questions

**Comments to the Author**

1. If the authors have adequately addressed your comments raised in a previous round of review and you feel that this manuscript is now acceptable for publication, you may indicate that here to bypass the “Comments to the Author” section, enter your conflict of interest statement in the “Confidential to Editor” section, and submit your "Accept" recommendation.

Reviewer #1: All comments have been addressed

Reviewer #2: (No Response)

2. Is the manuscript technically sound, and do the data support the conclusions?

Reviewer #1: Yes

Reviewer #2: Partly

3. Has the statistical analysis been performed appropriately and rigorously? 

Reviewer #1: Yes

Reviewer #2: Yes

4. Have the authors made all data underlying the findings in their manuscript fully available?

Reviewer #1: Yes

Reviewer #2: Yes

5. Is the manuscript presented in an intelligible fashion and written in standard English?

Reviewer #1: Yes

Reviewer #2: No

6. Review Comments to the Author

Reviewer #1: the language in submitted articles is clear, correct, and unambiguous. the statistical analysis has been performed appropriately and rigorously.

Reviewer #2: (No Response)

7. PLOS authors have the option to publish the peer review history of their article (what does this mean?). If published, this will include your full peer review and any attached files.

Reviewer #1: No

Reviewer #2: No

---

## [Author Response · Author response to Decision Letter 1]

15 Dec 2020

Response to review 

1. There are still major concerns about your article. Notably, a professional copy editor must be contacted to proof your manuscript for typographical errors and standard English.

Thank you for your suggestion. For addressing the issue we consulted senior public health staff and language professors in my university. We also used online software specifically grammerly and scribens (check for correctness of spellings).

---

## [Editor Report · Decision Letter 2]

17 Dec 2020

Magnitudes of post-abortion family planning utilization and associated factors among women who seek abortion service in Bahir Dar Town health facilities, Northwest Ethiopia, Facility-Based Cross-Sectional Study.

PONE-D-20-26289R2

Dear Dr. Bekele,

We’re pleased to inform you that your manuscript has been judged scientifically suitable for publication and will be formally accepted for publication once it meets all outstanding technical requirements.

Kind regards,

Frank T. Spradley

Academic Editor

PLOS ONE

---

## [Editor Report · Acceptance letter]

2 Jan 2021

PONE-D-20-26289R2 

Magnitudes of post-abortion family planning utilization and associated factors among women who seek abortion service in Bahir Dar Town health facilities, Northwest Ethiopia, Facility-Based Cross-Sectional Study. 

Dear Dr. Bekele:

I'm pleased to inform you that your manuscript has been deemed suitable for publication in PLOS ONE. Congratulations! Your manuscript is now with our production department. 

Kind regards, 

on behalf of

Dr. Frank T. Spradley 

Academic Editor

PLOS ONE